# Polyphenols, Saponins and Phytosterols in Lentils and Their Health Benefits: An Overview

**DOI:** 10.3390/ph15101225

**Published:** 2022-10-03

**Authors:** Ahmed M. Mustafa, Doaa Abouelenein, Laura Acquaticci, Laura Alessandroni, Simone Angeloni, Germana Borsetta, Giovanni Caprioli, Franks Kamgang Nzekoue, Gianni Sagratini, Sauro Vittori

**Affiliations:** School of Pharmacy, University of Camerino, Chemistry Interdisciplinary Project (CHIP) via Madonna delle Carceri, 62032 Camerino, Italy

**Keywords:** lentils, polyphenols, saponins, phytosterols, health-promoting effects

## Abstract

The lentil (*Lens culinaris* L.) is one of the most important legumes (family, Fabaceae). It is a natural functional food rich in many bioactive compounds, such as polyphenols, saponins and phytosterols. Several studies have demonstrated that the consumption of lentils is potentially important in reducing the incidence of a number of chronic diseases, due to their bioactive compounds. The most common polyphenols in lentils include phenolic acids, flavan-3-ol, flavonols, anthocyanidins, proanthocyanidins or condensed tannins, and anthocyanins, which play an important role in the prevention of several degenerative diseases in humans, due to their antioxidant activity. Furthermore, lentil polyphenols are reported to have antidiabetic, cardioprotective and anticancer activities. Lentil saponins are triterpene glycosides, mainly soyasaponins I and βg. These saponins have a plasma cholesterol-lowering effect in humans and are important in reducing the risk of many chronic diseases. Moreover, high levels of phytosterols have been reported in lentils, especially in the seed coat, and β-sitosterol, campesterol, and stigmasterol are the most abundant. Beyond their hypocholesterolemic effect, phytosterols in lentils are known for their anti-inflammatory activity. In this review, the current information on the nutritional composition, bioactive compounds including polyphenols, saponins and phytosterols, and their associated health-promoting effects are discussed.

## 1. Introduction

The lentil (*Lens culinaris* L.; Fabaceae family) is an annual plant native to Western Asia and other parts of the world, including North America [1,2]. Lentils are one of the world’s oldest crops, and theyhave been used in a variety of cuisines around the world, particularly in the Mediterranean and Indian areas [2]. It is most recognized for its lens-shaped edible seed, which contains the most important dietary components, including macro- and micronutrients [1]. Lentils, like most legumes, have been gaining popularity for their nutritional significance in human diets, with nearly 60% carbohydrate and 25% protein as macronutrients, which are the primary components of lentils. They are particularly high in essential amino acids such as lysine and arginine. Lentils also contain dietary fiber, as well as minerals such as iron, folate, magnesium, and zinc [2]. Because of these considerations, lentils have long been recognized as a low-cost, high-quality alternative to animal proteins, earning the term “poor man’s meat” [3], and they are regarded as a whole food supply for persons suffering from micronutrient malnutrition. Canada is the world’s top lentil exporter, selling to more than 100 countries each year and accounting for over a quarter of global output. The large green “Laird” cultivar and the red lentil are the two most frequently produced lentil cultivars [3].

Lentils are available in a variety of colors, including brown, black, red, yellow, and green, depending on the cultivar and the seed coat and cotyledon composition (Figure 1). The cotyledon color, which can be yellow, red, or green, is primarily responsible for the color of dehulled seeds. The entire seed color is determined by the seed coat, which might be brown, green, red, or black. Flavan-3-ols, proanthocyanidins, and certain flavanols are more abundant in lentil seed coats. This reveals that lentil seed coats may be more beneficial for a healthy diet [1]. Lentils have been categorized as soft seed-coated legumes, which require less cooking time, with consequent lower nutrient losses than hard seed-coated pulses. Lentils are highly convenient for human consumption, due to their shorter cooking time (23–26 min) compared to most other pulses (>70 min) [2].

Many studies in recent years have revealed that pulses, including lentils, provide potential health benefits beyond meeting basic nutrient requirements for humans [3]. Lentil consumption has been linked to lower rates of several chronic diseases such as cardiovascular disease (CVD), diabetes, malignancies, coronary heart disease, degenerative disorders, and aging [1,3]. The high amount of phytochemicals, such as polyphenols, saponins, and phytosterols in pulse-based diets including lentils, is responsible for some of these health benefits [1,3,4]. However, due to traditional eating habits, a lack of consumer education and knowledge of nutritional values, processing techniques, and available diversified food products, consumption of pulses such as lentils is limited in Western countries, with only approximately one in eight people consuming pulses on a daily basis. Despite this, Western customers are strongly advised to include pulses in their diets in order to receive the greatest health benefits [3]. Therefore, theFood and Agriculture Organization(FAO) named 2016 the International Year of Pulses to raise public awareness of the nutritional benefits of pulses as part of sustainable food production targeted at food security and nutrition (http://www.fao.org/pulses-2016/en/, accessed on 16 October 2021) [5].

Because of their high nutritional value and availability of bioactive secondary metabolites, scientists are increasingly interested in studying lentils as a functional food. The bioactive metabolites in lentils are important in the prevention of degenerative diseases in humans, as well as in boosting health. The present comprehensive review, which is based on exploratory investigations, aims to provide updated information on lentil bioactive components such as polyphenols, saponins, and phytosterols, as well as their health-promoting properties.

## 2. Lentils and Sustainability

Lentils belong to the legume family, and were used in agricultural practices in ancient history, when the oldest civilizations, such as Egyptians and Romans, used leguminous plants for green manuring and to enrich the soil. Nowadays, legumes represent the second source of human and animal consumption, after cereals [6]. On a quantitative basis, at the global level, lentils are predominant in import and export commodities and increased in production in the last decade. For instance, when analyzing the world import and export variations in tonnes for lentils during a 10-year time span (2009–2019), one can estimate a growth of +116.7% in imports and +159.9% in exports (import: 1,739,997 tonnes (2009) and 3,770,182 tonnes (2019); export: 1,500,458 tonnes (2009) and 3,899,913 tonnes (2019) (elaboration on FAOSTAT data, 2021). This positive trend is due mainly to a growing need for their use for human consumption, but also for animal breeding.

Studies have demonstrated that lentils are essential for a healthy diet, since they are rich in proteins, containing twice the protein amount as in corn and three times more than in rice, and they prevent obesity. In general, legumes, and thus lentils, are a valuable source of vegetable proteins (20–45%), with essential amino acids, complex carbohydrates (±60%) and dietary fibers (5–37%), and provide essential minerals and vitamins [6]; thus, their impact on the environment is modest compared with animal proteins, which result from breeding. Lentils are also low in fatty acids, free of cholesterol and gluten and rich in essential minerals and vitamins [7]. From a socio-economic perspective, they are a valuable source for human protein intake, much cheaper than animal protein and thus more accessible to lower-income families worldwide, and are key for global food security. In addition, lentils can be defined as a healthy food for soils, since they are rich in nutrients, are a source of income for millions of family farmers who cultivate them in alternation with other crops, and are essential for the replenishment of soil with nitrogen, improving the sustainability of production [8].

Due to the valuable functions of legumes, to which lentils belong, the United Nations proclaimed the International Year of Pulses in 2016, as cited above, for their key role in food and nutrition security, human health and soil, and adaptation to climate change. Lentils are being used frequently for soil erosion control, as a vegetal cover crop in strips alternating with clean-tilled crops, and in grass-legume mixtures on steep land retired from cultivation (FAO, 2019). Growing attention is being given to legumes, due to their important role as a soil nitrogen fixer. For mitigation and adaptation to climate change, lentils are plants capable of fixing atmospheric nitrogen into ammonia and converting it to enriched soil, while guaranteeing the biodiversity abundance in soils and preventing from erosion and arid fields. This helps in reducing the use of synthetic fertilizers, whose manufacture involves intensive energy consumption, with the consequent emissions of greenhouse gases into the atmosphere [9].

## 3. Polyphenol Constituents in Lentils

Lentils have the highest total phenolic content in comparison with other common legumes, such as green pea, chickpea, cowpea, yellow pea, mung bean and peanut. In addition to various known phenolic compounds, edible legumes, including lentils, mainly contain phenolic acids, flavonoids and condensed tannins. These compounds are distributed differently, where flavonoids are the main compounds of the seed coat, and non-flavonoids, such as hydroxycinnamic acids and hydroxybenzoic acids, are the main compounds of the cotyledon [10]. In general, low-pigment varieties have lower levels of phenolic compounds (e.g., anthocyanins) than high-pigment varieties. Thirty-five phenolic compounds have been reported in lentils, including catechins, procyanidins, flavanols, flavones, flavanone, hydroxybenzoic acids and hydroxycinnamic acids [1,11]. Caprioli et al. (2018) analyzed 16 polyphenols in four different types of lentils which can be considered a good source of polyphenols with an average content of 99.2 mg kg^−1^ [12]. Phenolic content reduced by 80%, 16–41% and 22–42% in lentils after decortication, cooking, and soaking, respectively [11].

Many methods, such as maceration and ultrasound-assisted extraction are available for phenolic compound extraction, and these methods are mainly affected by different factors (e.g., sample particle size, solvent, solute-solvent ratio, agitation, temperature, etc.). For the extraction of phenolic compounds, pure organic solvents (ethanol, methanol, acetone, ethyl acetate, etc.) alone or combined with water are often used. It was reported that aqueous acetone (70%, *v*/*v*) was the best solvent for phenolic compound extraction, followed by hot water, water, and aqueous ethanol (80%, *v*/*v*), and that the red lentil hull showed the highest content of phenolic compounds compared to the green lentil seed and hull [11]. HPLC DAD (High Performance Liquid Chromatography Diode Array Detector) and HPLC-MS (Liquid chromatography–mass spectrometry) are commonly used to elucidate the phenolic profile of plant species. These methods are ideal for the separation and quantification of phenolic compounds [13]. Spectrophotometric methods such as Folin-Ciocalteu are also used for determining the total phenolic content of different samples. However, being economic, simple, and quick methods, they are non-selective and cannot be used in individual phenolic compound quantification [13,14]. 

### 3.1. Phenolic Acids

Phenolic acids are compounds containing a phenolic ring attached to at least one carboxylic acid group. They differ in the number and position of the hydroxyl groups on the aromatic ring. Depending on the number of carbon units of the side chain attached to the phenolic ring, they can be divided into C6-C3, C6-C2, and C6-C1 compounds, the most important of which are C6-C3 (hydroxycinnamic acid derivatives) and C6-C1 (hydroxybenzoic derivatives). Hydroxybenzoic acid derivatives include gallic, vanillic, p-hydroxybenzoic, protocatechuic and syringic acid. However, hydroxycinnamic acid derivatives include p-coumaric, trans-ferulic, caffeic, sinapic and chlorogenic acids.

Raw lentils are reported to contain hydroxybenzoic acids: p-hydroxybenzoic and protocatechuic acid and hydroxycinnamic acids such as trans-p-coumaroyl malic, trans-p-coumaric, trans-p-coumaroyl glycolic, trans-ferulic, and cis-p-coumaric acids [11]. In another study on red lentil acetone extract, Amarowicz, et al. [15] reported the presence of p-hydroxybenzoic acid and gallic aldehyde as major hydroxybenzoic acids, while the major hydroxycinnamic acids were trans-ferulic, trans-p-coumaric and sinapic acid. A study on 11 lentil cultivars showed the presence of phenolic acid compounds of the benzoic and cinnamic types, namely gallic acid, protocatechuic acid, 2,3,4-trihydroxybenzoic acid, protocatechualdehyde, p-hydroxybenzoic acid and vanillic acid. Sinapic and chlorogenic acid were the main cinnamic acids; furthermore, caffeic acid, p-coumaric acid, syringaldehyde, m-coumaric acid and ferulic acid were identified [16]. In this study, the total content of benzoic and cinnamic acid derivatives were from 169 to 248 μg/g DW and from 1315 to 2381 μg/g DW, respectively. In green lentils, trans-p-coumaric acid, trans-ferulic acid and trans-p-coumaric acid derivative were the major phenolic acids [17]. The total content of hydroxybenzoic acids and hydroxy cinnamic acids in raw Pardina lentils was reported to be 5.69 and 3.76 μg/g, respectively. In the same variety, the major detected phenolic acids were dihydroxybenzoic acid, p-hydroxybenzoic acid, trans-p-coumaric, cis-p-coumaric and hydroxy acids [18]. In a study on 20 Canadian lentil cultivars (green and red lentils), trans-p-coumaric acid and p-hydroxybenzoic acid were the main detected phenolic acids [19]. Alshikh, et al. [20] also identified three benzoic acids in different lentil cultivars: gallic acid, methyl vanillate and protocatechuic acid, plus four hydroxycinnamic acids: caffeic, ferulic, p-coumaric and sinapic. A study carried out on four lentil cultivars using HPLC-ESI-MS^n^ analysis identifiedand quantifiedthe individual insoluble-bound phenolics upon boiling; it showed that the content of protocatechuic acid derivative decreased (from 28 to 49%) upon boiling, in all samples [21]. Another study [22] revealed the change of phenolic compounds during solid-state fermentation of three lentil cultivars and showed the formation of new phenolic compounds (resorcinol and cinnamic acid) in the *Aspergillus*-fermented lentil. However, vanillin and gallic acid, which are originally present in lentil extracts, were not observed in their fermented counterparts. This study by Liu, et al. [23] is in agreement with the results of Yeo and Shahidi [21], who reported the reduction of the sum of soluble and insoluble phenolic fractions after the boiling of lentils. In this study, the effects of four different cooking methods (microwave, slow, boiling, pressure) and three heating solutions (salt, sugar, water) on free (p-hydroxybenzoic, p-coumaric, ferulic and sinapic) and bound (p-coumaric and ferulic) phenolic acids was studied. Ghumman, et al. [24] studied the effect of lentil sprouting on different phenolic compounds, and the identified phenolic acids were gallic, chlorogenic, ferulic, sinapic, p-coumaric and caffeic acids. Sprouting was reported to have significant effect on quantity and composition of phenolic compounds, resulting in content increase of most of the phenolic compoundsin up to 48 h of sprouting; however, prolonged sprouting led to a reduction in content. This may be due to the increase in phenolic metabolism and enzyme action. In addition, soluble phenolic acids were tentatively identified in the hulls of four lentil cultivars (feruloyl glucoside, caffeoyl glucoside, coumaroyl glucoside 4-hydroxybenzoic acid glucoside, and 5-hydroxyferuloyl glucoside). Moreover, protocatechuic acid was detected as an insoluble phenolic acid in lentil hulls, and is present in the cell wall matrix [25]. Recently, the mean values of phenolic acids in the studied lentil genotypes were found to be 7.64 mg/100 g. The detected phenolic acids were 4-hydroxybenzoic acid as the main phenolic acid, followed by gallic acid, protocatechuic acid, p-coumaric acid and vanillic acid [26]. Giusti et al. (2018) showed that phenolic acids, namely gallic acid, caffeic acid, syringic acid and ferulic acid, were mainly significantly higher in the organic samples, with respect to the conventional ones [27].

### 3.2. Flavonoids

Flavonoids are the major phenolic compounds in legumes. They are composed of two aromatic rings (“A” and “B”) linked by a 3-carbon bridge (C6–C3–C6). Based on the structure of the flavonoids, they can be divided into six major classes: flavonols, flavanones, flavan-3-ols, flavones, isoflavones, and anthocyanins. Catechins (flavan-3-ols) and proanthocyanins(PAs, also known as condensed tannins, oligomers of catechins and epicatechins) are primarily present in legumes having colored seed coats. Another type of flavonoids is the anthocyanidin (aglycon);when it is bound to a sugar moiety, it is referred to as an anthocyanin. Anthocyanins are also mainly found in the colored seed coat of legumes [11].

Catechins, procyanidins and prodelphinidins are common in raw lentil samples, and generally show a decrease after enzymatic treatments. (+)-Catechin 3-O glucose is the major compound in raw lentil, followed by Prodelphinidin trimer (2GC-C). On the other hand, quercetin 3-O-rutinoside and myricetin 3-O-rhamnoside were reported to be the main flavonol glycosides. Apigenin 7-O-glucoside and apigenin 7-O-apiofuranosyl glucoside are the main flavone glycosides identified in raw lentils [11]. In red lentil extracts, Amarowicz, et al. (2009) [15] have detected and quantified the major flavonoids: prodelphinidin dimers, epicatechin, procyanidin trimers, digallate procyanidin dimer, procyanidin dimers, catechin glucoside, kaempferol derivative, catechin, procyanidin gallate, quercetin hexose acylated. In addition, epicatechin, catechin and luteolin have been detected in 11 lentil cultivars. Moreover, French green and Pardina lentils are reported to have an anthocyanin content of 157.3 and 665.6 μg/g delphinidin-3-glucoside equivalent, respectively [16]. Twenty bioactive compounds belonging to different chemical classes, dihydro flavonols, flavonols, hydroxycinnamates, dihydrochalcones, procyanidins and gallates were identified in the green lentils extracts, and catechin glucoside, quercetin diglycoside, procyanidin dimers and epicatechin glucoside were the major catechin derivatives. However, quercetin diglycoside was identified as the dominant flavone; other identified flavonols were acylated quercetin hexose and kaempferol glucoside [17]. As reported by Aguilera et al. (2010) [18], in raw lentils catechins and procyanidins constitute 69% of the total content of phenolic compounds (74.48 μg/g). The major detected ones were (+)-catechin 3-glucoside and procyanidin trimer. On the other hand, flavonols and dihydroflavonols constitute about 17% of the total identified phenolics, with kaempferol glycosides as the main flavonols. Kaempferol 3-rutinoside was the major flavonol, with luteolin 3′-7-diglucoside being the main flavone [18]. In green and red lentils, the content of anthocyanins was reported to be 3–4 fold higher in the hulls than in the seeds [11]. Mirali et al. (2014) [28] used the HPLC-MS method for identifying phenolic compounds present in the black, green and grey seed coats of three lentil genotypes. The contents of oligomeric flavan-3-ols were very similar in all genotypes, but luteolin and its glycosylated form were present in the highest content in the genotype with a black seed coat. As previously reported [20], the flavonoid content was 84 and 2000 folds higher than that found for hydroxycinnamic acids and hydroxybenzoic acids. In this study, flavan-3-ol monomers (catechin, epicatechin and (+)-catechin 3-glucoside), proanthocyanidins (procyanidin trimers A & C1, procyanidin dimers A & B and prodelphinidin dimer A), flavonol (kaempferol dirutinoside) and flavone (Luteolin 3′-7-diglucoside) were quantified in lentil cultivars [20]. Catechin was the major detected compound in the raw seeds of four lentil cultivars, and accounted for approximately 70% of total insoluble bound phenolics (IBPs). Other identified compounds in the IBPS were procyanidin dimer B2, epicatechin, catechin derivatives and carboxylated quercetin [21]. The contents of quercetin and catechin were found to be higher in *Aspergillus*-fermented lentils than their control counterparts [22]. A study on the effect of sprouting in different bioactive compounds reported an increase in catechin content up to 48 h after sprouting; however, longer sprouting led to a decrease in acid and base hydrolysed fractions. In addition, luteolin extracted in the base hydrolysed fraction increased with sprouting. On the other hand, luteolin extracted in the acid hydrolysed fraction decreased after prolonged sprouting [24]. Soluble and insoluble flavonoids have been detected and identified in lentil hulls of different cultivars. In the Greenland cultivar, catechin glucoside, prodelphinidin dimer, procyanidin dimer and trimer and quercetin-O-pentoside were the major phenolic compounds. Catechin glucoside showed the highest content in Greenland hulls; on the other hand, prodelphinidin dimer and quercetin were the major compounds in cultivar 3494–6 hulls. The hulls of cultivar Invincible contain a high content of catechin and catechin glucoside. The major soluble phenolics identified in hulls of the Maxim cultivar were prodelphinidin dimer, procyanidin dimer, catechin glucoside and procyanidin trimer [25]. Three flavonoids have been identified and quantified in 34 lentil cultivars: kaempferol-deoxyhexoside-hexoside I, kaempferol-deoxy hexoside-hexoside II and kaempferol-O-hexoside-O-deoxyhexo side-hexoside [29]. Recently, flavanols, their monomers and oligomers were also reported to be the major phenolics in lentils, where the mean values of flavanols across all lentil genotypes were 83.45 mg/100 g. The major flavanol in extracts of lentils was procyanidin dimer I (43.83 mg/100 g),which represented 53% of all identified phenolics, followed by gallocatechin, catechin, procyanidin B2 and catechin gallate [26]. Bubelova et al. (2018) reported that germination caused an increase in total phenolic and flavonoid contents and antioxidant activity, while cooking resulted in a reduction of all of these [30]. In addition, Djabali et al. (2020) indicated that the polyphenol content of lentils was reduced by heating [31]. Furthermore, Hard cap espresso machines were shown to be an alternative to currently used extraction methods for the easy, fast, and low cost analysis of lentil polyphenols, and the polyphenolic compound content was in the range of 4.6–5.5 mg per g lentils [32]. Total flavonoid content (TFC), condensed tannins content (CTC) and total phenolic content (TPC) in lentils are reported in Table 1, while various individual polyphenols are shown in Table 2.

## 4. Saponin Constituents in Lentils

The term saponin derives from the Latin word sapo, which means soap. In fact, these secondary metabolites possess emulsifying and foaming properties, since they are constituted of a lipophilic portion coupled with a hydrophilic part [37]. Not many years ago, saponins were considered antinutritional factors, due to their hemolytic, membranolytic and fungitoxic activities; after additional investigations and in-depth analyses, this definition has been almost abandoned [38]. On the contrary, at the present time saponins are known as bioactive compounds with interesting health properties such as anti-inflammatory, anticancer, cholesterol-lowering and prebiotic characteristics [39,40,41]. Saponins occur in more than 100 plant families, including Fabaceae, and in a few marine living organisms [37]. The present section will describe the chemistry of saponins, focusing on those present in pulses, especially in lentils. In addition, it will report the saponin contents and the common analytical approaches for their analysis in lentils.

### 4.1. Saponins in Lentils

Lentils, together with other legumes such as broad beans and kidney beans, are the main source of saponins in the human diet [37]. Lentils contain mainly triterpenoid glycosides; in particular, group A and B saponins and soyasaponin I (Bb), along with soyasaponin βg (VI) are the most representative compounds of this class [41]. Both are monodesmosidic saponins, with sapogenol B as aglycone, except for soyasaponin βg (VI), containing a 2,3-dihydro-2,5-dihydroxy-6-methyl-4H-pyran-4-one (DDMP) group at C-22, which generates major radical scavenger activity against reactive oxygen species (ROS) [42]. In addition, this saponin is considered a natural precursor of soyasaponin I, since the DDMP group is thermolabile [43]. Besides the common saponins reported in lentils, soyasaponin II (Bc) and Bd have been identified in 80% methanol extracts of lentils seeds by Ha et al. [44], while Herrera et al. [45] also identified a group A saponin along with soyasaponin I and two other group B saponins in dried extract of lentil seeds.

The saponin content in lentils varies according to the cultivars, locations, soil type, environmental conditions and time of cultivation, as well as processing techniques (soaking, sprouting, dehulling, cooking, blanching etc.) and method of extraction and analysis [38]. For instance, Sagratini et al. [42] have analyzed the contents of soyasaponin I and βg in 32 samples of Italian lentils (*L. culinaris*) with different testa colors and sizes (red, black, brown, and green; giant and mignon), and from diverse areas of cultivation. They found a total content of soyasaponins ranging from 180 to 1595 mg/kg, with soyasaponin βg (110–1242 mg/kg) being present at a higher concentration than soyasaponin I (28–407 mg/kg). Their findings also demonstrated that lentils with small-sized seeds had a higher content of saponins than those with bigger-sized seeds, while no correlation has been found between soyasaponin quantity and diverse testa color. In addition, they found that altitude influenced the content of these molecules in lentils. In fact, those cultivated at sea level or in the proximity of sea lands at low altitude showed la lower content of soyasaponins than those that grew in uplands. In another study, Donat et al. [43] found that lentils cultivated in fields at an intermediate altitude (1142–1387 m) showed the highest levels of soyasaponins. The contents of sapogenol A and B were studied in ten diverse genotypes of lentils by Srivastava and Vasishtha [46]. Sapogenol B (Figure 2) was present at a higher concentration than sapogenol A in all tested genotypes, with an average of 3775 and 2674 mg/kg, respectively. The contents of soyasaponin I and βg were tested in different raw and soaked and cooked legumes, including lentils, by Sagratini et al. [47]. They found that the total content of soyasaponins remained almost constant after soaking and cooking lentils (for example, lentils from Colfiorito resulted in 2542 and 2530 mg/kg for raw and cooked, respectively), but soyasaponin βg was partially converted into soyasaponin I. Moreover, a small part of these detergent compounds, i.e., 5.6–7.6%, leaked into the cooking solution. Similar findings were reported by an older pioneering study by Savarino et al. [48], who found a conversion of soyasaponin βg into soyasaponin I after cooking processes. Moreover, they found that the conversion into soyasaponin I increased with cooking time.

### 4.2. Quantitative Analysis of Saponins in Lentils

The saponin content in plants and, more specifically, in legumes is also influenced by the analytical process used, and has generally been extracted using different types of processes. These can be divided into conventional and green extraction technologies. The conventional extraction methods such as maceration, Soxhlet and reflux extraction, are based on the solubility of analytes into selected solvents, and therefore they often require a large amount of solvents. In contrast, green extraction techniques such as ultrasound-assisted extraction (UAE), microwave-assisted extraction (MAE) and accelerated solvent extraction (ASE), involve a lower solvent amount and safer chemicals, together with lower pollution [49]. The most common solvents used for extracting saponins are methanol, ethanol and a mixture of these with water [43,45,50,51,52]. Water and ethanol are preferred as extraction solvents, due to more ecofriendly features [49]. For the analysis of the aglycone part, after an extraction procedure acidic hydrolysis is necessary to hydrolyze saponin extract into sapogenin [45]. Once the lentil extract is prepared, the contents of saponins have been mainly studied by spectrophotometric and chromatographic methods. The spectrophotometric methods are simple, fast, and cheap and these are the reasons why they have become more and more popular for saponin quantitation in plant materials. The spectrophotometric method of election is the so-called vanillin-sulfuric acid assay. The selected wavelength is usually 544 nm [49]. For example, Ahuja et al. [53] estimated the total content of saponins in 44 lentil genotypes, finding a concentration that ranged from 1.70 to 3.50 mg/g, with an average value of 2.81 mg/g. A higher total saponin content (TSC) was found in dried extracts of lentils prepared by UAE using three different solvents, i.e., water, ethanol and ethanol:water (1:1, *v*/*v*) by Del Hierro et al. [51]. They found the highest value of TSC in ethanol extract (10.63 g/100 g) followed by ethanol:water (3.28 g/100 g) and water extract (0.19 g/100 g). The spectrophotometric approach gives only an estimation of the total saponin content, while for specific contents of each saponin, chromatographic methods are required. The usual chromatographic methods are high-performance liquid chromatography, coupled with mass spectrometry and/or diode array detector (HPLC-MS-DAD) and gas chromatography-mass spectrometry (GC-MS), which is the older approach limited to analyzing only the aglycones of saponins [54]. Before GC-MS analysis, the analytes must be derivatized, with bis(trimethylsilyl) trifluoroacetamide (BSTFA), for example, to permit the analysis in a gas system. One of the first HPLC analytical methods, developed to analyze triterpenoid glycosides such as soyasaponin I and βg, was that used by Savarino et al. [48]. They quantified these two saponins in raw, soaked, and cooked seeds of chickpeas and lentils using the HPLC-UV system. Subsequently, the same researchers have proposed the quantification of these two saponins by fast atom bombardment-mass spectrometry (FAB-MS) after extracting them with a Soxhlet apparatus from 20 cultivars of *L. culinaris* [55]. In 2009, Sagratini et al. [42] proposed an innovative analytical method to quantify two group B saponins in 32 samples of Italian lentil seeds. The analytical procedure was characterized by solid-liquid extraction with 70% aqueous ethanol, a purification step by solid-phase extraction (SPE) and HPLC-MS-DAD analysis. Some years later, Donat et al. [43] optimized a fast and simple method for soyasaponin I and βg quantitation in Italian lentils, avoiding the purification step and employing an HPLC-MS/MS system, which permitted the enhancement of specificity and sensitivity, with respect to an instrument with a single mass analyzer.

## 5. Phytosterols Constituents in Lentils

Plant sterols/stanols, also known as phytosterols (PS), are sterols of plant origin, and are important constituents of plant cell membranes, with established health benefits [56]. Their structure is similar to cholesterol, with a typicalcyclopentane-perhydrophe- nanthrene ring system, an -OH group at position 3 and a side chain at position 17 (Figure 3) [57].

### 5.1. Analysis of PS in Lentils

The extraction of PS is generally performed through a prior step of lipid extraction followed by the hydrolysis of the lipid samples for the separation of PS [58]. The extraction of lipids in legumes can be performed through conventional (Soxhlet, maceration) and non-conventional (ultrasound, microwave, subcritical extractions) extraction methods. The hydrolysis of fats is mainly performed with methanolic KOH through hot-saponification, to allow the extraction of the unsaponifiable fraction, mainly composed of PS. Other methods propose the direct extraction of PS in legumes, through the saponification of the legume powder [59], and the validation of phytosterol analysis by alkaline hydrolysis and trimethylsilyl derivatization, coupled with gas chromatography for rice products [59]. The hydrolysis step is also important to free the conjugated PS in legumes and thus analyze the total content of PS. Indeed, PS are present in plant cell membranes in a free form as well as in conjugated forms such as steryl esters (SE), steryl glycosides (SG) and acylated steryl glycosides (ASG). Acidic hydrolysis is fundamental to cleaving the glycosidic bonds in SG and ASG, while ester bonds in SE and ASG can be cleaved with both alkaline and acid hydrolysis. Once extracted, PS analysis is mainly performed by gas chromatography(GC-FID or GC–MS) and high-performance liquid chromatography (HPLC-DAD, HPLC-MS). The analyses of PS by GC requires a derivatization step through silylation with trimethylsiloxy groups (TMS), which allows a good separation and detection of PS [60]. However, GC requires high temperatures, which impact the stability sterols [61]. PS analysis with HPLC also presents some obstacles, due to the structural similarities of PS, which are difficult to separate and generally coelute. Moreover, due to their lack of chromophores and their high lipophilicity, PS have a low UV absorption intensity and cannot be detected using electrospray ionization (ESI-MS) [62]. Higher sensitivity can be obtained using other ionization techniques, such as atmospheric pressure chemical ionization (APCI) and atmospheric pressure photo ionization (APPI) [63]. Plant-based beverages are good sources of free and glycosidic plant sterols [63,64]. A rapid method for the simultaneous quantification of the major tocopherols, carotenoids, free and esterified sterols in canola (Brassica napus) oil using normal phase liquid chromatography [64]. Moreover, chemical derivatization can be performed before HPLC analysis for the derivatization of PS using reagents such as dansyl chloride (DNSCl), which could increase the UV and MS detection of PS [65]. The development of an innovative phytosterol derivatization method to improve HPLC-DAD analysis, and the ESI-MS detection of plant sterols/stanols is also reported [65].

### 5.2. PS Levels in Lentils

As in other pulses, PS in legumes are mainly from the 4-desmethyl sterols subgroup with β-sitosterol, campesterol, stigmasterol, and Δ^5^-avenasterol as principal compounds (98%) [66,67]. Minor levels of plant stanols are also present, with saturated rings and side chains (β-sitostanol, campestanol, and cycloartenol) [68]. These main PS compounds occur in free and conjugated forms. Conjugated PS are significant in lentils and include not only sterol esters (fatty acid esters and hydroxycinnamic acid esters) but predominantly high levels of sterol glycosides and acylated sterol glycosides, which represent between 10 and 30% of total PS [68,69].

High levels of PS have been reported in lentils; in fact, the seedcoat of lentils is a naturally rich source of PS [45]. Indeed, the content of PS in lentils ranges from 20 to 160 mg/100 g. β-Sitosterol is the most abundant PS in lentils, representing between 65 and 80% of the total PS [67,68,70]. β-sitosterol (15–125 mg/100, g) campesterol (2–15 mg/100 g) and stigmasterol (2–20 mg/100 g) are the main contributors to the PS profile of lentils (Table 3).

The PS content in lentils can be reduced by prolonged heating processes and hrough their oxidation, leading to the formation of phytosterols oxidation products (POPs). Major POPs include 7α/β-hydroxysterols, 7-ketosterols, 5α,6α-/5β,6β-epoxysterols and 3β,5α,6β-triols, which demonstrate cytotoxic and pro-inflammatory effects [56]. However, to date, no study has reported the content of POPs in lentils.

## 6. Health Benefits of Lentils Polyphenols

The increasing interest in studying natural products and food containing them is related to their potential health benefits. In this field, one of the most common class of natural compound studied is that of polyphenols, because of their potential role in human health [71,72], which makes them ‘lifespan essential’. Biological activities of lentil polyphenols were studied [1], and were reported to have potential health benefits, such as antidiabetic, antioxidant, cardioprotective and anticancer activities. The phenolic compounds in lentils, mainly flavonols, are good inhibitors of α-glucosidase and lipase enzymes, which are associated with the digestion of lipids and glucose in the intestine, and so contribute significantly to the control of blood sugar and obesity. Consumption of lentils has a potential role in weight management and blood sugar control [19]. Moreover, lentils can be used as a prognostic marker for various cancers, including thyroid and hepatic carcinoma. 

### 6.1. Anti-Diabetic Activity

Studies demonstrated that the regular consumption of lentils can be useful for the prevention and management of diabetes [73], thanks to their ability to improve blood glucose and lipid and lipoprotein metabolism [74]. Other studies highlight the role of flavonoids and fibers in lentils in gut motility and in the prevention of impairment of metabolic control in diabetic rats, suggesting their potential role in diabetic patients [1]. Moreover, the regular consumption of cooked lentils (50 g) among diabetic patients leads to a reduction in fasting blood sugar (FBS), glycemic load and glycemic index in steptozotocin (STZ)-induced diabetic animals [75,76]. This is due to the activity of polyphenols in lentils, which have a health-promoting impact on metabolic disorders such as diabetes, obesity, coronary heart disease and CVD [77]. In vivo and in vitro studies also demonstrate the role of lentils in the regulation of starch digestibility, glycemic load, and glycemic index, with a subsequent decrease in diabetes complication [78]. In conclusion, the regular consumption of lentils is effective in the prevention and management of diabetes.

### 6.2. Anti-Oxidant Activity

Lentils have different groups of phenolic compounds acting as antioxidants, due to their ability to reduce the formation of reactive oxygen species, such as the superoxide anion, by chelating metal ions or inhibiting enzymes [79,80]. Among these, the most important are procyanidin and prodelphinidin dimers and trimers, gallate procyanidins, kaempferol derivatives, quercetin glucoside acetate, luteolin derivatives and p-coumaric acid, hydroxybenzoic compounds, protocatechuic, vanillic acid, aldehyde p-hydroxybenzoic, trans-ferulic acid and trans-p-coumaric acid.

### 6.3. Anti-Obesity Activity

Several studies demonstrate the role of lentils in controlling obesity. Among these, epidemiological studies demonstrate the influence of lentil polyphenols in reducing the incidence of obesity and diabetes. A human study also showed the connection between the intake of lentil seed with pasta and sauce and the reduction of food intake, body weight and waist circumference [81]. Moreover, observational studies assess the inverse connection between the consumption of lentils and the basal metabolic index or risk associated with obesity [82]. Finally, interventional studies demonstrate the potential of lentils to inhibit α-glucosidase and pancreatic lipase, with subsequent reduction of glucose and fat digestion and absorption in the intestine. Moreover, the ability of lentil polyphenols to control postprandial glucose and fat, which is important for the management of diabetes and obesity, has been demonstrated [83].

### 6.4. Cardioprotective Effect

Polyphenols in lentils have a potential role in reducing the occurrence of various cardiovascular diseases (CVDs). Indeed, polyphenols reduce blood pressure by inhibiting the angiotensin I-converting enzyme [84]. Moreover, lentil polyphenols have antihyperlipidemic, hypohomocysteinemic, anti-cholesterolemic and cardioprotective effects, with subsequent reduction of the risk of hypertension and coronary artery disease [85]. In vivo studies on hypertensive animal models have demonstrated that consuming lentils reduces total cholesterol (TC), triglycerides (TG), low density lipoprotein (LDL) and the pathological manifestation of cardio-morphometric analysis. This suggests the possible use of lentil seeds as therapeutic agents in hypertensive patients [86]. The consumption of lentil seeds reduces the glycemic index and hyperlipidemic effects in diabetic animal models. This study demonstrated that lentils increase the high density lipoprotein (HDL) levels and reduces blood glucose levels [76]. In conclusion, the regular consumption of lentils reduces the risk of cardiovascular and coronary artery diseases.

### 6.5. Anticancer Activity

Lentil seeds reduce the influence of colon, thyroid, liver, breast and prostate cancers [80,87]. An epidemiological study on 90,630 women demonstrates that the consumption of polyphenol-rich lentils reduces the risk of breast cancer. The chemo-preventive activities of lentil polyphenols include the uptake of carcinogens, activation or formation, detoxification, binding to DNA and fidelity of DNA repair [88]. Lentil seeds have a chemo-preventive potential on colorectal carcinogenesis; this has been demonstrated using azoxymethane, which reduces the number of dysplastic lesions and neoplasms in the colon of rats [88,89].

The health benefits of lentil polyphenols are summarized in Table 4.

## 7. Health Benefits of Saponins

As previously reported, lentils, as with pulses in general, have been recommended for inclusion in the daily diet, to control or prevent the risk of chronic and degenerative diseases, by many researchers and health organizations [101]. Among the bioactive components of lentils, saponins are a much-studied class of compounds. The structural complexity of this class of compounds is responsible for their physical, chemical, and biological properties, fundamental for their activities and applications. Saponin structure is characterized by one hydrophilic part (glycone) and one lipophilic part (sapogenin). This structure allows them, in aqueous solution, to align with the lipophilic part away from water, decreasing the surface tension and causing foaming [102].

Saponins are a very diverse group and only a few of these properties are common to all of them. Collecting information on saponin biological properties and health benefits is of great value. Numerous researchers have focused on saponin biological activities. Recently, many epidemiological and experimental studies have shown saponins’ beneficial effects on human health. Several sources of evidence suggest that saponins not only seem to have antioxidant and cholesterol-lowering capacities, but also anti-inflammatory, antibacterial, antidiabetic, and neuro-protective activities, as well as stereoselective effects on ion channel current regulation and the cardiovascular system, immunostimulation, and protection against the risk of cancer development [39,40,103,104].

Consuming a diet rich in legumes containing saponins can also reduce the risk of heart disease and control plasma cholesterol in humans [41]. In addition, the oral intake of soya saponin has been used as an adjuvant aid in the treatment of retroviral infection and to prevent obesity [105,106]. Many studies are focusing on their mechanism within, and applications to, dietary supplement development [107,108].

### 7.1. Antioxidant Properties, Hypocholesterolemic Effect and Gut Microbiota Health Impact

The oxidation mechanism involves an attack on lipids by oxygen radicals which can remove doubly allyl hydrogen from polyunsaturated lipids and promote lipid peroxidation. Saponins are considered as a natural antioxidant, since they can bind cholesterol and prevents its oxidation in the colon [38]. Saponins show antioxidant activity in different ways. Some of them can capture free radicals, while some others can activate antioxidant enzymes to inhibit the formation of complexes between free radicals and metallic ions. Luo et al. (2016) tested the antioxidant capacity of saponins extracted from legumes through total flavonoid and phenolic contents, radical DPPH scavenging activity and ferric reducing antioxidant power assay. They concluded that saponins, mainly concentrated in the hull of legumes, have a very high antioxidant activity [109].

The hypocholesterolemic mechanisms of saponins follow differentroutes. Some saponins can interact directly with cholesterol and form an insoluble saponin–cholesterol complex, which inhibits cholesterol absorption from the small intestine. Other saponins lower cholesterol by increasing fecal excretion of bile acids which solubilize cholesterol in the small intestine [105]. A hydroalcoholic extract of lentils was tested by Micioni di Bonaventura et al. (2017) to examine the hypocholesterolemic action on an animal model, by studying bile acids concentration in feces and the potential prebiotic effect. Lentil extract reduced rat cholesterol levels by16.8% and increased bile acids level in rat feces, revealing the same prebiotic activity of inulin and good bifidogenic activity [41].

Saponin-rich extracts from red lentils (*Lens culinaris*) were tested by Del Hierro et al. (2019) in in vitro colonic fermentation, to evaluate their impact on human fecal microbiota. After lentil extract fermentation, 4 μg/mL of sapogenin was found in the form of soyasapogenol B. Lentil extracts showed antimicrobial effect, mainly on lactic acid bacteria and *Lactobacillus* spp. They concluded that saponins in lentil extracts can be transformed into sapogenins by human gut microbiota, with a modulatory effect on the growth of selected intestinal bacteria [105].

Antioxidant-rich foods such as lentils and other legumes can decrease starch and lipid digestions that are fundamental in diabetes management [110]. Xie et al. (2018) concluded that saponins can also reduce pro-inflammatory cytokines/mediators in serum, liver, and white adipose tissues, improve serum lipid profiles, decrease liver triglyceride and cholesterol, and promote fecal and biliar excretion [111].

### 7.2. Anti-Inflammatory Activities

Saponins from lentils were also studied as anti-inflammatory agents [104]. Conti et al. (2021) investigated the impact of a legume diet on elderly people and underlined how soyasaponins from lentils and other legumes can exhibit anti-inflammatory activity. These bioactive compounds can inhibit the production of pro-inflammatory cytokines (TNF-α and MCP-1, PGE2 and NO), inflammatory enzymes (COX-2 and iNOS) and the degradation of IκB-α (an inhibitor of NF-κB, in LPS-stimulated macrophages) [40]. It was demonstrated that saponin extracts from legumes revealed a dose-dependent inhibitory activity against inflammation. Studies on the antiallergic activity of saponin constituents from pulses are also available [109].

### 7.3. Inhibition of Cancer

According to the American Institute for Cancer Research (AICR), legumes bioactive phytochemicals may protect against cancer [112]. Saponins can interact with free- or membrane-bound sterols in the colon and stimulate their fecal elimination. This mechanism could be strictly correlated with saponin cancer inhibition potential. Several studies report that these could be valid mechanisms for the anticarcinogenic effects of saponins [107,108]. Faris et al. (2020) reviewed epidemiological evidence supporting the impact of lentil seeds and their bioactive compounds on lowering cancer risk. Estimated lentil saponin content is 34 mg/100 g. Saponins in lentils are expressed as soya sapogenol B. Extracted lentil saponins have been reported as possible antitumor agents [39]. Various health benefits of lentil saponins are summarized in Table 5.

## 8. Health Benefits of Phytosterols in Lentils

The most studied effect and the main reason for global interest in PS is their cholesterol-lowering effect [113]. Indeed, PS in lentils have been shown to lower/reduce blood cholesterol with a daily intake of 1.5–2.4 g. This effect could be explained through four main mechanisms:in the intestinal lumen, PS esters compete with cholesterol esters for hydrolysis by digestive enzymes;being structurally analogous and more lipophilic than cholesterol, PS compete with cholesterol, decreasing its micellar solubility in the intestine; the non-integrated cholesterol into mixed micelles will thus be non-solubilized and the cholesterol absorption by enterocytes will be strongly reduced [114]. In enterocytes, PS can inhibit ACAT-2, interrupting the esterification of cholesterol; the unesterified cholesterol will be excreted from enterocytes into the intestinal lumen by cholesterol transporter proteins [115]. With regard to cholesterol transporters, three transporter proteins regulate cholesterol absorption: Niemann–Pick C1-Like 1 transporter (NPC1L1) allows the cholesterol to enter into enterocytes, while ATP-binding cassette (ABC) G5/G8 transporters (ABCG5 and ABCG8), are responsible for its efflux from enterocytes and its secretion from hepatocytes into bile [116]; PS enhance the expression of the genes encoding ABCG5 and ABCG/8 transporters [117]; moreover, PS compete with cholesterol for the NPC1L1 transporter but are simultaneously poorly absorbed (1–15% of absorption) [118].

Additionally, PS can decrease the biosynthesis of cholesterol by reducing the expression of the HMGCS1 gene. Indeed, cholesterol de novo biosynthesis is catalyzed by HMG-CoA synthase 1, which is encoded by HMGCS1 [119].

Beyond their hypocholesterolemic effect, PS in lentils are known for their anti-inflammatory [2,3] and anti-tumor properties [120]. Indeed, PS are reported to prevent and protect against common cancer types such as prostate, breast, colon, and other cancers [121]. Invitro studies, giving a supplementation of β-sitosterol 16 μmol/L in a human tumor cell line for five days exhibited an inhibitory effect on tumor growth of HT-29 cells, a human colon cancer tumor cell line [122]. This effect is explained by their ability to alter the cell membrane structure of cancerous cells, impacting their membrane integrity and fluidity [39].

## 9. Conclusions

Lentils are considered to be one of the most nutritious and health-beneficial food known. They are sustainable food sources that have the potential to alleviate global food shortages, which will be a major challenge in the near future. Taking this into account, incorporating lentil seeds into people’s diets all across the world may help to alleviate obesity and malnutrition. According to the present review of the phytochemical composition and biological activities, lentils contain a variety of health-promoting bioactives, such as polyphenols, saponins, and phytosterols, which have been shown to be beneficial in the management and prevention of a variety of human chronic diseases. These compounds have antioxidant properties and have a key role in preventing diseases such as diabetes, obesity, CVD, and cancer. The development of lentil-based functional food products and nutraceuticals should be widely promoted, due to their nutritional and health-promoting potential. Furthermore, more research is needed to improve and optimize agricultural and culinary circumstances so that the abundant amount of bioactive phytochemicals in lentils can be fully utilized.

## Figures and Tables

**Figure 1 pharmaceuticals-15-01225-f001:**
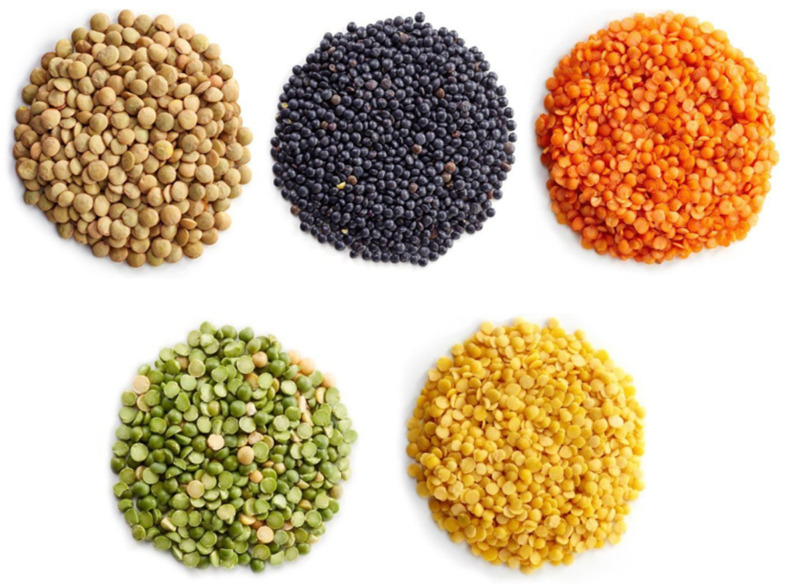
Different types of lentils.

**Figure 2 pharmaceuticals-15-01225-f002:**
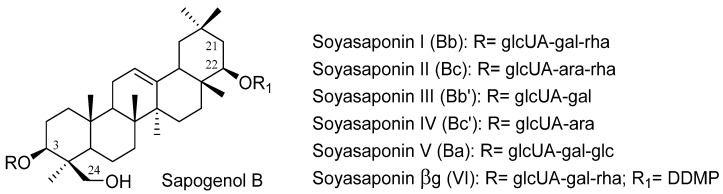
Group B saponins including the main DMMP saponin present in lentils.

**Figure 3 pharmaceuticals-15-01225-f003:**
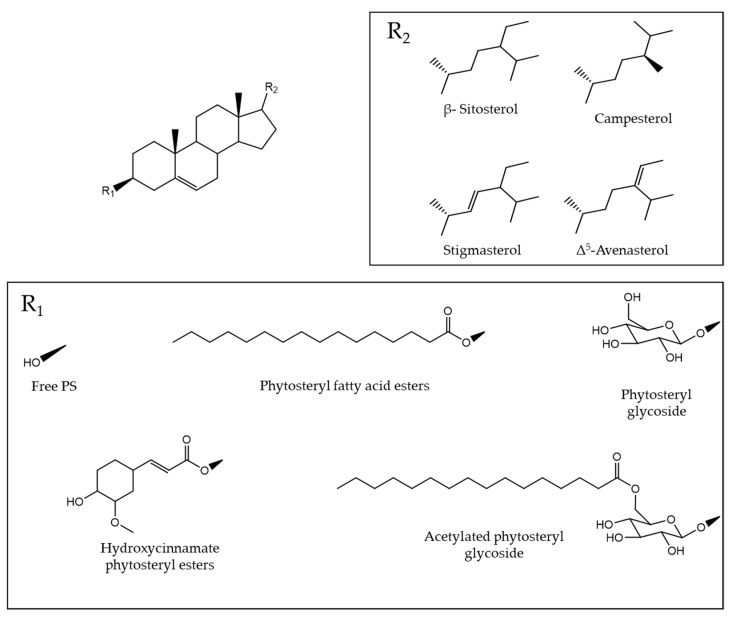
Chemical structure of the main phytosterols in legumes.

**Table 1 pharmaceuticals-15-01225-t001:** Total flavonoid content (TFC), condensed tannin content (CTC) and total phenolic content (TPC) reported in lentils.

Sample	TFC	CTC	TPC	
Lentil varieties	3.04 to 4.54 mg CE/g DW	3.73 to 10.20 mg CE/g DW	4.9 to 7.8 mg GAE/g DW	[11]
Red Chief lentil from Spokane	2.21 mg CE/g DW	-	7.53 mg GAE/g DW	[11]
Lentil varieties	-	-	12 mg GAE/g DW	[11]
Red Lentil	-	70 A_500_/g	58 mg CE/g crude extract	[15]
Lentil (11 cultivars)	2840 to 6870 μg/g DW	-	1543 to 2551 μg/g DW	[16]
Green lentil	-	93 A_500_/g	68 mg CE/g DW	[17]
Morton lentils	30.0 mg CE/g DW	61.6 mg CE/g DW	70.0 mg GAE/g DW	[33]
Green lentil seed and hull	-	-	Entire: 5.0 to 19.3 Hull: 24.6 to 82.9 mg CE/g DW	[11]
Red lentil seed and hull	-	-	Entire: 5.1 to 20.8 Hull: 27.2 to 87.2 mg CE/g DW	
Lentils (3 cultivars)	-	269.0–37.8 mg/100 g FW.	-	[34]
Lentil	-		47.6 mg GAE/g DW	[35]
Green lentil (10 cultivars)	0.7 to 1.9 mg CE/g DW	3.4 to 7.8 mg CE/g DW	4.6 to 8.3 mg GAE/g DW	[19]
Red lentil (10 cultivars)	0.6 to 1.6 mg CE/g DW	3.0 to 5.8 mg CE/g DW	4.6 to 7.6 mg GAE/g DW	
Lentil cultivars	Free: 0.01 to 0.8 mg CE/g. Esterified: 0.4–4.1 mg CE/g. Insoluble bound: 0.08–2.95 mg CE/g	Free: 0.4 to 2.7 mg CE/g. Esterified: 0.6–10.5 mg CE/g. Insoluble bound: 0.03–4.3 mg CE/g	Free: 1.4 to 5.5 mg GAE/g. Esterified: 2.3–21.5 mg GAE/g. Insoluble bound: 2.6–17.5 mg GAE/g	[20]
Lentil cultivars	-Before boiling: 3.9 to 5.4 mg CE/g.-After boiling: 3.5 to 4.4 mg CE/g	-	-Before boiling: 6.2 to 7.9mg GAE/g. -After -After Boiling: 5.5 to 6.6 mg GAE/g	[21]
Lentil seeds	8.43 mg quercetin/g	-	5.21 mg GAE/g	[36]
Three lentil cultivars	-	1.56 to 2.40 mg CE/g DW	15.8 to 17.5 mg GAE/g DW	[22]
Lentils	-	-	4.13 mg ferulic acid equivalents/g DW	[23]
Lentil hulls of four different varieties			soluble phenolics in lentil hulls (10.71–45.85 mg/g) insoluble- bound phenolics in lentil hulls (6.71–4.18 mg/g)	[25]
Pardina variety (34 samples)	-	-	30 to 150 μg/100 g FW	[29]
5 genotypes of lentil at 10 different locations	4.34 to 5.31 mg CE/g DW	2.67 to 3.98 mg GAE/g DW	5.62 to 6.38 mg GAE/g DW	[26]

Catechin equivalent (CE), Gallic acid equivalent (GAE), Dry weight (DW), FW (Fresh weight).

**Table 2 pharmaceuticals-15-01225-t002:** Phenolic compounds reported in different types of lentils and their quantities.

Chemical Class	Chemical Sub-Class	Phenolic Compounds	Quantity	References
Phenolic acids	Hydroxybenzoic acids	Gallic acid	90.9–136.8, 15.5 8 μg/g	[11,16,20,22,24,26,27]
		Protocatechuic acid or Dihydroxybenzoic acid	1.45, 3.68, 20.2–37.7 μg/g	[11,16,18,19,20,26]
		p-Hydroxybenzoic acid	1.90, 3.25, 73.46, 15.7–44.9, 22.8, 3.62–5.80 μg/g DW (red lentils), 2.93–5.74 μg/g DW (green lentils).	[11,15,16,18,19,23,26]
		2,3,4 Trihydroxy benzoic	16.9–29.2 μg/g	[16]
		Syringic acid		[19,27]
		Vanillic acid	0.59–3.22 μg/g	[16,26]
	Hydroxycinnamic acids	trans-p-Coumaric acid	5.74, 38.84, 37.3, 2.14, 4.24–11.19 μg/g DW (red lentils), 4.70–12.94 μg/g DW (green lentils)	[11,15,16,17,18,19,20,22,23,24,26]
		cis-p-Coumaric acid	0.73 μg/g	[11,16,18]
		m-Coumaric acid		[16]
		trans-Ferulic acid	0.74, 15.99, 10.1 μg/g	[11,15,16,17,20,23,24,27]
		trans-p-Coumaroyl-malic acid	10.02 μg/g	[11,18,19]
		trans-p-Coumaroyl-glycolic acid	2.88 μg/g	[11,18]
		Sinapic acid	0.06, 1099–2217 μg/g	[15,16,20,23,24]
		Chlorogenic acid	159–213 μg/g	[16,24]
		Caffeic acid	-	[16,20,27]
		Cinnamic acid	-	[22]
			-	
Flavonoids	Flavones and flavonols	Quercetin 3-O-rutinoside	-	[11]
		Apigenin hydrate	-	[25]
		Apigenin 7-O apiofuranosyl glucoside	6.20 μg/g	[11]
		Apigenin 7-O glucoside	1.87 μg/g	[11]
		Apigenin hexose	-	[15]
		Quercetin	3300 μg/g	[17,22,25]
		Quercetin diglycoside or Quercetin diglucoside	114 μg/g	[15,25]
		Quercetin hexose	-	[15]
		acylated quercetin hexose	37.2 μg/g	[17]
		Quercetin-O-pentoside	-	[25]
		Dihydroquercetin	-	[17]
		Quercetin-3-O-glucoside	-	[17,19,25,28]
		Quercetin-3-O-galactoside,	-	[17,28]
		Quercetin-3-O-rutinoside	5.24 μg/g	[17]
		Carboxylated quercetin	-	[21]
		Kaempferol	-	[25]
		Kaempferol-3-O-rutinoside	5.95 μg/g	[17,18]
		Kaempferol -dirutinoside	-	[18,20]
		Kaempferol derivative	-	[15]
		Kaempferol 3-glucoside,	19.4 μg/g	[18,19,25,28]
		Dihydrokaempferol hydrate	-	[25]
		Dihydrokaempferol glycoside	-	[18]
		Dihydrokaempferol dimer	-	[25]
		Kaempferol acetylglycoside	-	[18]
		Kaempferol–rhamnoside hexose-hexose	-	[18]
		Kaempferol 3-robinoside-7-rhamnoside	-	[18]
			-	
		Kaempferol triglycoside	-	[19]
		Kaempferol tetraglycoside or Kaempferol tetraglucoside	-	[19,25]
		Kaempferol-deoxyhexoside-hexoside isomer I	23.0 to 157.8 g/100 g FW	[29]
		Kaempferol-deoxyhexoside-hexoside isomer II	0.0–157.8 g/100 g FW	[29]
		Kaempferol-O-hexoside-O-deoxyhexoside-hexoside	0.0–157.8 g/100 g FW	[29]
		Apigenin methyl ether	-	[18]
		Myricetin	-	[25]
		Myricetin 3-rhamnoside	5.95 μg/g	[11,18,28]
		Luteolin	18–77 μg/g	[11,24,25,28]
		Luteolin 3′-7-diglucoside	4.55 μg/g	[18,20]
		Luteolin-7-O-glucoside	-	[11,25,28]
		Luteolin-4′-O-glucoside	-	[25,28]
	Isoflavones	Formononetin rhamnoside	-	[25]

	Flavanones	Eriodictyol 7-rutinoside	-	[18]
		Eriodictyol 5,7-dimethoxyflavone	-	[18]
		Naringenin	-	[28]
		Naringenin glucoside	-	[25]

	Dihydrochalcones	Phloridzin	-	[15]

	Flavan-3-ols and procyanidins	Catechin	36.02, 2410, 267–1899 μg/g, 320–2160 μg/g DW	[15,16,17,18,20,21,22,24,25,26,28]
		Catechin-3-glucoside	31.50, 51.95, 289, 39.89, 6590, 2230, 2790 μg/g	[15,18,19,20,25]
		Gallocatechin	-	[26,28]
		Catechin gallate	-	[19,26]
		Epicatechin	98.21, 2535–4946 μg/g	[15,16,17,18,20,21,25]
		Epicatechin glucoside	59.5 μg/g	[15]
		Epicatechin gallate	-	[19]
		Procyanidin	-	[18,25]
		Procyanidin dimer	65.98, 100, 3240 μg/g	[15,18,19,21,25]
		Digallate procyanidin dimer	83.29 μg/g	[15]
		Procyanidin dimer A	-	[20]
		Procyanidin dimer B or procyanidin B2	-	[20,26]
		Procyanidin dimer I	438.3 μg/g	[26]
		Procyanidin trimer	87.5, 9.30, 2590 μg/g	[15,18,25]
		Procyanidin trimer A	-	[20]
		Procyanidin trimer C1	-	[20]
		Procyanidin gallate	32.78 μg/g	[15,25]

	Prodelphinidins	Prodelphinidin	-	
		Prodelphinidin dimer	161.6, 3340, 5800 μg/g	[15,20,25]
		Prodelphinidin trimer (2GC-C)	31.05 μg/g	[11,15]

	Anthocyanins	Delphinidin- 3-glucoside	-	[16]
		Malvidin-3-O-galactoside	-	[28]
Stilbenes	-	trans-Resveratrol-3-O-glucoside	-	[11]
		trans-Resveratrol.	-	[11]
Coumarins	-	4-Hydroxy-6-methylcoumarin	-	[28]

**Table 3 pharmaceuticals-15-01225-t003:** Levels of phytosterols in legumes expressed as mg of sterols/100 g of lentils.

Lentils	β-Sitosterol	Campesterol	Stigmasterol	Δ^5^Avenasterol	Stanols
L. green small ^1^	58.5 ± 3.3	7.0 ± 0.5	9.3 ± 0.6	-	3.7 ± 0.5
L. green large ^1^	62.5 ± 1.4	9.1 ± 0.6	8.3 ± 0.3	-	3.7 ± 0.3
L. orange ^1^	54.4 ± 2.4	7.7 ± 0.3	6.8 ± 0.2	-	3.0 ± 0.5
L. yellow ^1^	53.5 ± 3.1	4.2 ± 0.2	2.7 ± 0.2	-	2.1 ± 0.1
L. small Fakés ^2^	24.2 ± 1.0	2.1 ± 0.2	2.6 ± 0.1	2.5 ± 0.2	-
L. small ^3^	123.4 ± 4.1	15 ± 0.4	20 ± 0.6	-	-

^1^ Data retrieved from Nyströmet al. [68]. ^2^ Data retrieved from Kalogeropoulos et al. [67]. ^3^ Data retrieved from Ryan et al. [70].

**Table 4 pharmaceuticals-15-01225-t004:** Health benefits of polyphenols in lentils.

Health Benefits	Mechanism of Action	Ref.
Prevention and management of diabetes	Improves blood glucose, lipid and lipoprotein metabolism.	[73,74]
Anti-diabetic	Reduces fasting blood sugar (FBS), glycemic load and glycemic index in streptozocin (STZ)-induced diabetic animals.	[75,76,90,91]
Reduced diabetic complications	Regulates starch digestibility, glycemic load and glycemic index.	[78]
Anti-obesity	Controls post-prandial glucose and fat digestion.	[83]
Reduction of CVDs	Reduces the total cholesterol (TC), triglycerides (TG), low density lipoprotein (LDL) and pathological manifestation of cardio-morphometric analysis.	[86]
Reduce the glycemic index and hyperlipidemic effects in diabetic animal model	Increases the high density lipoprotein (HDL) levels and reduces blood glucose levels.	[76]
Chemo-preventive on colorectal carcinogenesis	Reduces the number of dysplastic lesions and neoplasms in the colon of rats.	[88,89]
Antioxidant	Reduces the formation of reactive oxygen species.	[79,80,92,93,94,95,96,97,98]
Reduction of CVDs	Reduces blood pressure by inhibiting angiotensin I-converting enzyme.	[84]
Reduced risk of hypertension and coronary artery disease	Antihyperlipidemic, hypohomocysteinemic, anti-cholesterolemic effects.	[85]
Chemo-preventive, anticancer	Uptake of carcinogens, activation or formation, detoxification, binding to DNA and fidelity of DNA repair.	[88,99]
Gut motility and potential role in diabetic patients	Prevents the impairment of metabolic control in diabetic rats.	[1]
Oxidative Stress-Induced Hepatotoxicity	Lentil phenolic extract protects liver cells against oxidative stress, partly by inducing cellular antioxidant system; thus, it has hepatoprotective effects	[100]

**Table 5 pharmaceuticals-15-01225-t005:** Health benefits of lentil saponins.

Biological Activity of Saponins	Main Findings	Reference
Gut microbiota health	Saponins in lentil extracts can be transformed into sapogenins by human gut microbiota, with a modulatory effect on the growth of selected intestinal bacteria.	[105]
Anticancer	Estimated lentil saponin content is 34 mg/100 g. Extracted lentil saponins have been reported as possible antitumor agents. Saponins in lentils are expressed as soya sapogenol B.	[39]
Anti-inflammatory	Soyasaponins from lentils and other legumes inhibit the production of pro-inflammatory cytokines (TNF-α and MCP-1, PGE2 and NO), inflammatory enzymes (COX-2 and iNOS) and the degradation of IκB-α (an inhibitor of NF-κB, in LPS-stimulated macrophages).	[40]
Hypocholesterolemic and prebiotic	Ethanol/water saponin-rich extracts from lentils can reduce the cholesterol level of rats by 16.8% and increase bile acid levels in the feces of rats. Tested lentils extract shows the same prebiotic activity of inulin and good bifidogenic activity.	[41]

## Data Availability

Data sharing not applicable.

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
