# Peer review of "Polyphenols, Saponins and Phytosterols in Lentils and Their Health Benefits: An Overview"

_pharmaceuticals, 2022, doi:10.3390/ph15101225_

Round 1
Reviewer 1 Report (Previous Reviewer 3)
The last changes were implemented.
Reviewer 2 Report (Previous Reviewer 2)
The manuscript entitled Polyphenols, Saponins and Phytosterols in Lentils and Their Health Benefits: an Overview can be accepted for publication.
Reviewer 3 Report (New Reviewer)
The topic of this review is interesting. There are some errors regarding the way of writing the chemical names of compounds.
A- Careful revision of the chemistry section is required
B- Please correct as follows throughout the text:
1- para, ortho, meta or its abbreviations p, o and m should be in italics.
2- The letter O for oxygen in the compounds names such as Luteolin-7-O-glucoside and other examples, should be italic.
3- cis and trans should be in italic.
4- Careful revision of the chemical names, Table 2. Apigenin hexose, quercetin hexose and others, it should be hexosides.
5- Kaempferol–rhamnoside hexose-hexose, what does this refer to in table 2?
6- I suggest drawing a summary for the structures of phenolic compounds and flavonoids as the authors did for saponins.
7- Title of table 3: espressed or expressed?
8- Lines 567-576: please add the references after the respective sentences not as a group at the end of the paragraph.
9- For references, as per the journal guidelines, abbreviations of journal names are required in the reference list. please adjust.
10- Latin names of plants must be in italic, please revise throughout the text and reference list.
This manuscript is a resubmission of an earlier submission. The following is a list of the peer review reports and author responses from that submission.
Round 1
Reviewer 1 Report
I think this is a well written paper.
Author Response
R: Thank you very much to the reviewer for the very positive feedback.
Reviewer 2 Report
The paper of Mustafa et al. is a very nice review on the main bioactive compounds found in lentils. The information is presented in a nice manner and is rather well structured.
The authors were very generous with the details regarding the saponins, but information regarding the methods used for polyphenols and phytosterols are scarce. I suggest presenting, in short, those methods, such as to provide the reader with a complete picture. This addition would better balance the content of sections 3. Polyphenol Constituents in Lentils, 4. Saponin Constituents in Lentils and 5. Phytosterols Constituents in Lentils.
Author Response
The paper of Mustafa et al. is a very nice review on the main bioactive compounds found in lentils. The information is presented in a nice manner and is rather well structured.
R: Thank you very much to the reviewer for the very positive feedback.
The authors were very generous with the details regarding the saponins, but information regarding the methods used for polyphenols and phytosterols are scarce. I suggest presenting, in short, those methods, such as to provide the reader with a complete picture. This addition would better balance the content of sections 3. Polyphenol Constituents in Lentils, 4. Saponin Constituents in Lentils and 5. Phytosterols Constituents in Lentils.
R: Thank you to the reviewer for the kind request. We added, in short, those methods of polyphenols (lines 140-151), and phytosterols (lines 418-456) according to the reviewer suggestion.
Reviewer 3 Report
The manuscript “Polyphenols, Saponins and Phytosterols in Lentils and Their Health Benefits: an Overview” was submitted to Pharmaceuticals for publication.
Broad comments:
The review article gives a good overview on the phytochemistry of lentils and the health-beneficial effects on the discussed compound classes. It is in most parts very well written and contains a lot of data, which in my opinion needs to be structured differently to make the data easier accessible for the reader.
Such an example is Table 1, where main findings and lists of compounds are depicted in single lines, with the latter being repeated several times. Here, it would make more sense to list the compounds in a separate table and give the references, in which they were reported. The main findings should be removed from the table and be discussed in the text.
On the other hand, a lot of information is given in the body that would be better shown in a table, as e.g. the many reports of compound quantities in sections 3.1. and 3.2. Maybe, these values can also be depicted in the compound table (see above).
Another point, is that the manuscript is not always concise. From time to time, the authors move too much into basic science or chemistry, which then does not fit with the rest of the article. Section 1, in example, can be removed entirely, as is represents only basic knowledge. Moreover, the beginning of section 4.3. can be reduced and lines 450 to 456 can be deleted. Also, the part of section 7 before 7.1. seems like it has been already described before. Please check.
Also, most of the tables are not concise in their selves. Mixing lentils and compound classes, or the terms lentils and Lens culinaris Medik (as in Table 4) does look very scientific. The same accounts for the terms Phenolic compounds and Polyphenols or the combination of Flavonoids and fibers (both in table 3). The first two classes are not clearly diversified and the second two classes do not belong together at all.
Summarizing, the review article has its strengths, which is the collection and good writing, but also a clear deficit, which is the presentation of the data. That is most evident in the tables, which must be revised completely.
Another question is, if the authors chose the right journal for their review article. Here, MDPI offers, several other journals (e.g. Antioxidants, Molecules, Foods, Nutrients, Neutraceuticals) that would be more appropriate for the topic.
Specific comments:
Line 23: Please write “especially IN the seed coat”
Line 33: Please write “Fabaceae/Faboideae” instead of “Leguminose family”
Line 104: The authors write that lentils are low in cholesterol and gluten. Is this really the case, or are lentils free of cholesterol (no animals) and gluten (no cereals)?
Line 118: Please write “attention IS given”
Line 160: Please write “sinapic” in small letters
Line 164: Please insert a colon after lentils
Line 170: Please write “In a study on..”
Line 355: Please explain DDMP here, when deleting section 4.1.
Line 577: Please write “Antioxidant Properties…”
Author Response
The review article gives a good overview on the phytochemistry of lentils and the health-beneficial effects on the discussed compound classes. It is in most parts very well written and contains a lot of data, which in my opinion needs to be structured differently to make the data easier accessible for the reader. Such an example is Table 1, where main findings and lists of compounds are depicted in single lines, with the latter being repeated several times. Here, it would make more sense to list the compounds in a separate table and give the references, in which they were reported. The main findings should be removed from the table and be discussed in the text.
R: Thanks you to the reviewer for the kind comment. We did consider to structure the Table as suggested by the reviewer, but we opted for the present form because we think it gives more immediate info on the summary of cited references. With our approach the reader can have concise info, without having to search in the text.
On the other hand, a lot of information is given in the body that would be better shown in a table, as e.g. the many reports of compound quantities in sections 3.1. and 3.2. Maybe, these values can also be depicted in the compound table (see above).
R: Thanks to the reviewer for the kind comment. In these sections, given the fact that “a lot of information is given in the body”, we think that it would be quite difficult to put those info in a table without losing most of the information. For this reason we think it would be better to keep the table in the current form.
Another point, is that the manuscript is not always concise. From time to time, the authors move too much into basic science or chemistry, which then does not fit with the rest of the article. Section 4.1, in example, can be removed entirely, as is represents only basic knowledge. Moreover, the beginning of section 4.3. can be reduced and lines 450 to 456 can be deleted. Also, the part of section 7 before 7.1. seems like it has been already described before. Please check.
R: Thank you to the reviewer for the kind suggestion. Section 4.1 was removed, also the beginning of section 4.3. has been reduced (e.g. lines 411-414 were removed) and lines 450 to 456 were deleted, according to the reviewer suggestion. More, as suggested, the part of section 7 before 7.1 has been sensibly shortened.
Also, most of the tables are not concise in their selves. Mixing lentils and compound classes, or the terms lentils and Lens culinaris Medik (as in Table 4) does look very scientific. The same accounts for the terms Phenolic compounds and Polyphenols or the combination of Flavonoids and fibers (both in table 3). The first two classes are not clearly diversified and the second two classes do not belong together at all.
R: Thanks to the reviewer for his comment. Following this suggestion, we deleted the first column in Tables 3 and 4.
Summarizing, the review article has its strengths, which is the collection and good writing, but also a clear deficit, which is the presentation of the data. That is most evident in the tables, which must be revised completely.
R: Thanks to the reviewer for his comment. We revised the tables and almost all the recent article dealing with these topics were reported.
Another question is, if the authors chose the right journal for their review article. Here, MDPI offers, several other journals (e.g. Antioxidants, Molecules, Foods, Nutrients, Neutraceuticals) that would be more appropriate for the topic.
R: Thanks to the reviewer for his kind comment. We think that Pharmaceuticals journal is suitable for our review, also because we were invited to write the present paper after submitting an abstract of it.
Specific comments:
Line 23: Please write “especially IN the seed coat”
R: Suggestion has been implemented.
Line 33: Please write “Fabaceae/Faboideae” instead of “Leguminose family”
R: Suggestion has been implemented all over the manuscript.
Line 104: The authors write that lentils are low in cholesterol and gluten. Is this really the case, or are lentils free of cholesterol (no animals) and gluten (no cereals)?
R: As properly suggested, lentils are free of cholesterol and gluten. The mistake was corrected all over the manuscript.
Line 118: Please write “attention IS given”
R: Suggestion has been implemented.
Line 160: Please write “sinapic” in small letters
R: Suggestion has been implemented.
Line 164: Please insert a colon after lentils
R: Suggestion has been implemented.
Line 170: Please write “In a study on..”
R: Suggestion has been implemented.
Line 355: Please explain DDMP here, when deleting section 4.1.
R: Suggestion has been implemented.
Line 577: Please write “Antioxidant Properties…”
R: Suggestion has been implemented.
Reviewer 4 Report
I think that the manuscript entitled “Polyphenols, Saponins and Phytosterols in Lentils and Their Health Benefits: an Overview" deserves publication in the Pharmaceuticals after minor revision. The manuscript is clearly written and the lentil knowledge is systematized.
Line 93: please delete “.”
Line 104: lentils – “cholesterol”?
Line 135: please complete “Caprioli et al. […]”
Please standardize the names of the ingredients, incl. italic, capital letters. In all manuscript
Line 221: please change “Amarowicz, Estrella, Hernández, Dueñas, TroszyÅ„ska, KosiÅ„ska and Pegg” into “Amarowicz et al.”
Lines 233-234: please change “Aguilera, DUENas, Estrella, Hernandez, Benitez, Esteban and Martin-Cabrejas” into “Aguilera et al.”
Please correct the correctness of the punctuation marks throughout the manuscript
Please standardize „et al.” Throughout the manuscript according to the journal rules
Line 251: please delete „the”
Line 278: please complete „Bubelova et al. […]”
Line 280: please complete “Djabali et al. […]”
Line 501: please change “In vivo and in vitro” into “In vivo and in vitro” and in all manuscript
Line 584: please complete “Luo et al. […]”
Line 593: please complete “…Bonaventura et al. […]”
Line 597: please complete “Del Hierro et al. […]”
Lines 600-601: please change “Lactobacillus” into „Lactobacillus”
Line 605: please complete “Xie et al. […]”
Lines 611-612: please complete ” Conti et al. […]”
Line 626: please complete “Faris et al. […]”
Author Response
I think that the manuscript entitled “Polyphenols, Saponins and Phytosterols in Lentils and Their Health Benefits: an Overview" deserves publication in the Pharmaceuticals after minor revision. The manuscript is clearly written and the lentil knowledge is systematized.
R: Thank you very much to the reviewer for the very positive feedback.
Line 93: please delete “.”
R: Suggestion has been implemented.
Line 104: lentils – “cholesterol”?
R: As properly suggested, lentils are free of cholesterol (and gluten). The mistake was corrected all over the manuscript.
Line 135: please complete “Caprioli et al. […]”
R: Suggestion has been implemented.
Please standardize the names of the ingredients, incl. italic, capital letters. In all manuscript
R: Suggestion has been implemented.
Line 221: please change “Amarowicz, Estrella, Hernández, Dueñas, TroszyÅ„ska, KosiÅ„ska and Pegg” into “Amarowicz et al.”
R: Suggestion has been implemented.
Lines 233-234: please change “Aguilera, DUENas, Estrella, Hernandez, Benitez, Esteban and Martin-Cabrejas” into “Aguilera et al.”
R: Suggestion has been implemented.
Please correct the correctness of the punctuation marks throughout the manuscript
R: Suggestion has been implemented.
Please standardize „et al.” Throughout the manuscript according to the journal rules
R: Suggestion has been implemented.
Line 251: please delete „the”
R: Suggestion has been implemented.
Line 278: please complete „Bubelova et al. […]”
R: Suggestion has been implemented.
Line 280: please complete “Djabali et al. […]”
R: Suggestion has been implemented.
Line 501: please change “In vivo and in vitro” into “In vivo and in vitro” and in all manuscript
R: Suggestion has been implemented.
Line 584: please complete “Luo et al. […]”
R: Suggestion has been implemented.
Line 593: please complete “…Bonaventura et al. […]”
R: Suggestion has been implemented.
Line 597: please complete “Del Hierro et al. […]”
R: Suggestion has been implemented.
Lines 600-601: please change “Lactobacillus” into „Lactobacillus”
R: Suggestion has been implemented.
Line 605: please complete “Xie et al. […]”
R: Suggestion has been implemented.
Lines 611-612: please complete ” Conti et al. […]”
R: Suggestion has been implemented.
Line 626: please complete “Faris et al. […]”
R: Suggestion has been implemented.
Round 2
Reviewer 3 Report
Dear authors,
several minor points that were raised in the previous review have been improved. However, the two major points have not been addressed. The structure of table 1, which is going over 7 pages, does not help at all and in contrast to the authors' opinion it is everything but concise. Also giving numerous values in the text which would ideally suit for a table is not helpful. It rather looks as if the authors were not willing to do the work, which is a pity because it could have been a good review article.